# (Not That) Essential: A Scoping Review of Migrant Workers’ Access to Health Services and Social Protection during the COVID-19 Pandemic in Australia, Canada, and New Zealand

**DOI:** 10.3390/ijerph19052981

**Published:** 2022-03-03

**Authors:** Satrio Nindyo Istiko, Jo Durham, Lana Elliott

**Affiliations:** 1School of Public Health, Faculty of Medicine, The University of Queensland, Herston, QLD 4006, Australia; 2School of Public Health and Social Work, Faculty of Health, Queensland University of Technology, Kelvin Grove, QLD 4059, Australia; joanne.durham@qut.edu.au (J.D.); lana.elliott@qut.edu.au (L.E.); 3College of Public Health, Medical and Veterinary Sciences, James Cook University, Townsville, QLD 4810, Australia

**Keywords:** migrant workers, COVID-19, access, migrant farm workers, international students, social protection

## Abstract

Migrant workers have been disproportionately affected by the COVID-19 pandemic. To examine their access to health services and social protection during the pandemic, we conducted an exploratory scoping review on experiences of migrant workers in three countries with comparable immigration, health, and welfare policies: Australia, Canada, and New Zealand. After screening 961 peer-reviewed and grey literature sources, five studies were included. Using immigration status as a lens, we found that despite more inclusive policies in response to the pandemic, temporary migrant workers, especially migrant farm workers and international students, remained excluded from health services and social protection. Findings demonstrate that exploitative employment practices, precarity, and racism contribute to the continued exclusion of temporary migrant workers. The interplay between these factors, with structural racism at its core, reflect the colonial histories of these countries and their largely neoliberal approaches to immigration. To address this inequity, proactive action that recognizes and targets these structural determinants at play is essential.

## 1. Introduction

Migrant workers, defined by International Labour Organization (ILO) [1] as “international migrant individuals of working age and older who are either employed or unemployed in their current country of residence” (p. ix), continue to be disproportionally affected by the COVID-19 pandemic [2,3,4,5]. Accounting for 4.7% of the total global workforce, two-thirds of migrant workers reside in high-income countries [1,6], and are often employed in essential sectors, where physical distancing can be difficult and access to protective equipment limited [7,8], resulting in increased potential exposure to COVID-19 [9,10,11].

Lack of commitment from host countries to guarantee access to health promoting resources also exacerbates the vulnerability of migrant workers [12]. These health promoting resources are often embedded within health services and social protection schemes and include health care, employment, housing, and welfare support [13,14]. However, the World Health Organization (WHO) recently reviewed COVID-19-related policies globally and argued that high-income countries, such as Australia, Canada, and New Zealand, have enacted inclusive policies that should enable access to health services and social protection for all migrants, including migrant workers [15]. There is, hence, a need to investigate whether the promise of inclusive policies outlined in the literature is matched by the lived experience of migrant workers’ access to health services and social protection during the COVID-19 pandemic.

Access is influenced by immigration status, which is a product of immigration policies that provides legal status for migrants to reside in host countries [13,14,16]. During the COVID-19 pandemic, evidence indicates that host countries continue to use immigration status in their COVID-19-related policies to determine what and how diverse migrant populations can access health promoting resources [17]. Building upon this body of knowledge, examining countries with comparable immigration policies, where migrant workers are grouped based on similar immigration status, can provide a critical understanding of their access-related experiences. As such, using the lens of immigration status, this exploratory scoping review aims to examine access to health services and social protection among migrant workers in Australia, Canada, and New Zealand during the COVID-19 pandemic.

We begin by briefly comparing the immigration policies of Australia, Canada, and New Zealand to consider historical aspects of such policies and understand how their current policies categorize migrant workers [18]. Following this comparison, we provide an overview of COVID-19-related policies for migrant workers across these three settler societies before describing the methods and results of this review. Finally, we discuss the results and provide recommendations for the future research agenda.

### 1.1. Immigration Policies in Australia, Canada, and New Zealand

Australia, Canada, and New Zealand share similar colonial pasts, and in each of these settler societies, immigration has largely been used as a means of nation-building and driven by economic objectives [18,19]. Settler societies are defined as states established through colonization of Indigenous societies and the use of selective immigration policies, either based on racial or economic discrimination, to grow the nation’s wealth while maintaining a sense of national identity [18]. In migration studies, scholars often group Australia, Canada, New Zealand, and the United States of America together in a sub-grouping given the historic similarities in their immigration policies [18,19]. For the purpose of this health services and social protections-focused research, the United States of America was excluded given the nation’s distinctively different approach to these domains, as evidenced in a recent study by Hill, Rodriguez and McDaniel [17] on immigration status as a barrier to care in United States during the pandemic. While in-depth examination of immigration policies between Australia, Canada, and New Zealand is not the focus of this paper per se, in this section, we highlight key similarities in the way their immigration policies have evolved.

From the late nineteenth to mid-twentieth century, Australia, Canada, and New Zealand implemented immigration policies which discriminated non-European migrants based on colonial ideas of white supremacy [19,20]. The White Australia Policy provides a clear example of this colonial mentality [18,20]. Following World War II, however, there was a need for an increased labor force to rebuild the economy [19], which contributed to a shift in policy by prioritizing economic potential of migrants over social or racial factors [19,20].

In the early 1990s, the establishment of the European Union, which allowed for free movement of workers within the Union, created a shortage of skilled migration in other Western nations, including Australia, Canada, and New Zealand [20]. In response, immigration policy arrangements evolved to include different categories of migrants based on temporary and permanent immigration status [21]. This double-layered immigration policy has remained the primary strategy to attract and retain migrant workers [21]. Migrant workers with temporary immigration status, hereafter referred as ‘temporary migrant workers’, include international students, working holidaymakers, and seasonal workers [22]. They are at high risk of exploitation and tend to have fewer employee entitlements when compared to migrant workers with permanent immigration status (i.e., permanent migrant workers) [22]. As a result, many temporary migrant workers have become symbolic of low-waged, and low-skilled, but essential labor [23], continuing the interconnected forces of colonial histories and neoliberal economic agenda that shape immigration policies across the three settler societies [18,24].

During the COVID-19 pandemic, Australia, Canada, and New Zealand have continued to focus their immigration policies on meeting internal labor needs in essential industries [15]. In Australia, for example, the government has continued to facilitate entry for seasonal farm workers from countries with pre-existing bilateral agreements [15,25,26], extended the 40-h work per fortnight limitation for international students working in supermarkets [22], and offered a 12-month extension visa for other temporary migrant workers in essential and non-essential sectors (e.g., tourism and hospitality) [27]. Canada and New Zealand have employed similar short-term migration strategies [15].

### 1.2. COVID-19-Related Policies in Australia, Canada, and New Zealand

Since the onset of the pandemic, rapid policy changes have also occurred in health, welfare, and employment sectors [6,15,28]. WHO categorized these COVID-19-related policies as either aiming to provide short-term solutions to minimize the spread of COVID-19 (e.g., protective equipment in the workplace) or provide medium- and long-term solutions (e.g., subsidized or free access to health services and increased social protection) [15]. In Australia, Canada, and New Zealand, these reactive policies have altered the health services and social protection afforded to migrant workers, as outlined below.

In the context of health policies, Australia, Canada, and New Zealand have ensured long-term free access to COVID-19 testing and treatment for everyone regardless of their immigration status [15]. Each country has also invested in resources that support the translation and distribution of COVID-19 testing, treatment, and vaccine information to culturally and linguistically diverse populations [15,29]. Welfare measures, however, tend to focus on short-term solutions and have been more specifically targeted towards international students and permanent residents. These measures have focused mainly on support in finding alternative employment, access to cash payments, travel assistance, and emergency housing and food services [22,30,31].

In assessing government policies across the three countries, three major differences are of note. First, the Australian government was unique in excluding temporary migrant workers from accessing COVID-19 wage subsidy [22]. By contrast, the governments of Canada [32] and New Zealand [33] extended the eligibility criteria for COVID-19 wage subsidies to include temporary migrant workers. Second, in the context of employment policies, Canada was the only country that paid attention to medium- and long-term working conditions of migrant workers. For example, CAD 58.6 million was invested in supporting migrant farm workers with improved accommodation, farm inspections regimes, and workplace safety [34]. These actions were the result of a high profile COVID-19 outbreak among migrant farm workers in British Columbia in March 2020 [35]. Thirdly, the Canadian government granted permanent residency to a limited number of temporary migrant workers with specified positions in essential and certain non-essential industries (e.g., retail) [36]. This policy has also addressed migrant workers’ previous exclusion from a wide range of health services and social protection measures based on their immigration status; a measure proposed to be adopted in Australia [37] but not New Zealand at the time of the writing (January 2022).

## 2. Materials and Methods

This study was designed as an exploratory scoping review and followed the Joanna Briggs Institute’s scoping review methodology [38].

### 2.1. Database Search Strategy

Three databases (PubMed, Medline, and Scopus) were searched on 17 December 2021. The search strategy utilized the key terms ‘COVID’, ‘immigrant’, ‘risk’, and their derivatives as well as the countries of interest separated by Boolean operator ‘AND’. Given that ‘migrant workers’ and immigration status are often ill-defined [1,14], broader search terms were engineered to initially capture a wider cross-section of papers discussing the impacts of COVID-19 on migrant populations before delineating down. The full search strategy can be found in Table 1.

Database searches by L.E. returned 957 sources, yielding 931 original English papers once duplicates were removed. As outlined in Figure 1, articles were then excluded sequentially based on title, abstract and full text. S.N.I. and L.E. screened the articles utilizing the following criteria: (a) focused on migrant workers (defined as any migrants aged 15 years and older [1]); (b) included primary data on migrant workers’ experiences, barriers, and facilitators in accessing health services and/or social protection schemes during the COVID-19 pandemic; (c) focused geographically on Australia, Canada, and New Zealand; (d) published after 1 January 2020; and (e) written in English.

Title screening was undertaken by L.E. using EndNote reference manager software. The preliminary results of title screening were discussed with S.N.I. to reach consensus before 875 papers were removed, given they were not focused on migrant workers or based in the countries of interest. Subsequent abstract and full text screening was undertaken by S.N.I. and L.E. and facilitated through web-based software Rayyan [39] to ensure blinding. An additional 51 paper were removed through abstract screening, with many not meeting the country or population criteria or with content not focused on experiences of access to health services or social protection. Full text double screening by S.N.I. and L.E. saw the further removal of three articles primarily based on their insufficient focus on access to health and social protection. This left a final set of two peer-reviewed research papers from Caxaj and Cohen [40] and Farbenblum and Berg [41]. Reference lists of included sources, reviews, and excluded papers of interest were also searched by hand but did not result in the inclusion of additional papers. Any discrepancies throughout the screening process were resolved through consensus discussions between S.N.I. and L.E.

### 2.2. Grey Literature Search Strategy

Acknowledging both the recency of changes in this research space and the important role of government and non-governmental organizations (NGOs) in supporting migrant communities [12,34], S.N.I. searched through selected agencies in each country. In Australia, the websites of Australian Government Department of Health, Department of Home Affairs, Migrant Workers Justice Initiative, and Monash Migration and Inclusion Centre were searched. In Canada, the agencies that we selected were Health Canada, Employment and Social Development Canada, Immigration, Refugees, and Citizenship Canada (IRCC), World Education Services, and Migrant Workers Alliance for Change. Lastly, we chose the Ministry of Health, Immigration New Zealand, and Belong Aotearoa as the key agencies in New Zealand. These sources were identified through a Google search and a discussion among the authors. A snowballing approach was applied to ensure the sector was sufficiently canvased.

The grey literature search strategy applied the same key words and inclusion criteria as those used in the database search. The search was undertaken by S.N.I. in consultation with L.E. and drew on publication repositories and search functions found on each organizations’ website. Google site searches utilizing the government domains for each country were also undertaken. All reviewed publications from government agencies were policy announcements and did not provide primary data, and thus, they were excluded. An academic report [22] identified through the grey literature search and a paper found through the database search [41] are based on the same study by Berg and Farbenblum. With their report providing a more comprehensive and detailed account of findings [21], the peer-reviewed publication was excluded as a duplication [35]. This left one report from academic actors [22], two NGO reports (Migrant Workers Alliance for Change [42] and Belong Aotearoa [43]), and one report from a commercial actor (World Education Services [44]). The inclusion of four grey literature publications reflects the centrality of the role of non-government actors, particularly NGOs, in supporting migrant workers. A bibliographic search of the included reports was also conducted, but did not yield the inclusion of additional sources.

### 2.3. Data Extraction and Synthesis

A final set of five peer-reviewed papers and reports [22,40,42,43,44] were obtained for data synthesis through the combined database and grey literature search strategies. Data extraction and synthesis was undertaken collaboratively and utilizing an iterative approach. S.N.I. and L.E. conducted the data charting and collation using Google Spreadsheet. Extracted data were mapped collaboratively to identify paper characteristics (author, year, country, aim, study design and methodology, and dates of data collection), respective study populations (size, immigration status, industry of employment, and demographic information), and key themes relevant to the aim of this review. Demographics of migrant workers included in each source were initially examined. Immigration status was subsequently used to identify key sub-groups of migrant workers, permitting a narrative synthesis of their experiences in accessing health services and social protection. Critical insights into data charting and analysis were also sought from J.D.

## 3. Results

The results indicate that despite being two years into the COVID-19 pandemic, Australian, Canadian, and New Zealand-based empirical evidence exploring access to health services and social protection among the migrant workers remain limited. In stark contrast to the exponential growth in COVID-19-related research more broadly, just five papers fit the selection criteria for this study including three focused on migrant workers in Canada [40,42,44], one in Australia [22], and one in New Zealand [43]. Three studies drew on survey data [22,43,44], one used interview and focus group discussions [40], and one was based on calls made to a support hotline for migrant workers [42].

### 3.1. Demographics of Included Migrant Workers

Two of the Canada-based papers focused on temporary migrant workers working in farms (hereafter—‘migrant farm workers’) [40,42]. Caxaj and Cohen [40] explored how migrant farm workers accessed support services through group discussions and interviews with support workers linked to this community. Meanwhile, Migrant Workers Alliance for Change [42] used discussions through a support hotline to document COVID-19-related issues experienced by migrant farm workers. Given these study types, delineating demographics data of migrant farm workers was not possible in this review (see Table 2). Demographics information, however, was obtained from the remaining three studies, demonstrating a concentration of female, migrant workers aged 25 and above, who were born in Asia, and had lived in the host countries for at least 12 months (see Table 2) [22,43,44].

### 3.2. Immigration Status of Included Migrant Workers and Their Employment Sectors

Table 2 above also provides data on the immigration status of migrant workers across the included studies. Based on Table 2, two Canada-based papers focused explicitly migrant farm workers [40,42]. Meanwhile, World Education Services [44] and Belong Aotearoa [43] included temporary and permanent migrant workers in Canada and New Zealand, respectively, with the latter including those who gained citizenship. Lastly, Berg and Farbenblum [22] focused solely on temporary migrant workers in Australia.

Among the sources, two sub-groups of temporary migrant workers were of particular focus: migrant farm workers [40,42] and international students [22,43,44]. While migrant farm workers’ immigration status is tied to their employment in agriculture sector, international students are employed across essential (e.g., health and social services) [44] and non-essential sectors (e.g., hospitality) [22,44]. Both groups experienced exclusion from health services and social protection, contributing to their COVID-19 vulnerability [22,40,42]. We provide in-depth descriptions of their experiences with access to health services and social protection in the following sections.

### 3.3. Access to Health Services during the COVID-19 Pandemic

Most migrant farm workers in Canada live in accommodation provided by their employers within or near their farm of employment [40,42]. These locations often present an increased risk of COVID-19 due to poor ventilation and crowded spaces, which contributed to a farm-based COVID-19 outbreak in March 2020 [40]. Many migrant farm workers who acquired COVID-19 were often blamed for failing to access testing in a timely manner, yet in order to access COVID-19 testing and treatment, they typically had to rely on the benevolence of their employers [40]. Unfortunately, exploitative employment practices within the agricultural sector created fear of job loss and deportation among COVID-19-positive migrant farm workers, resulting in many choosing not to report their symptoms to their employers [40].

International students’ access to COVID-19 testing and treatment has received less attention across the included sources. Instead, the studies reported that many international students in Australia [22] and Canada [44] have lost their jobs or had reduced work hours, further pushing them into precarity, which is defined as “a multidimensional construct encompassing dimensions such as employment insecurity, individualized bargaining relations between workers and employers, low wages and economic deprivation, limited workplace rights and social protection, and powerlessness to exercise workplace rights” [45] (p. 230). Their precarious employment created financial difficulties to cover cost of consultation or essential medicine for non-COVID-19 medical conditions, despite having private health insurance as mandated by their visa [22]. In contrast, Belong Aotearoa [43] reported that international students and other migrant workers in New Zealand have not experienced the same level of job or income loss. However, they reported that female temporary migrant workers were more likely to lose their jobs or had their hours reduced.

The included studies further highlight the role of NGOs as a source of information and practical support to navigate access to health services related to COVID-19 and non-COVID-19 medical conditions [42,43]. As a result, NGOs, along with Facebook and ethnic groups, become the primary source of COVID-19 information for many migrant worker communities [43].

### 3.4. Access to Social Protection during the COVID-19 Pandemic

Prior to the Canadian government’s significant investment to improve the living and working standards of migrant farm workers in July 2020 [34], many were exploited by their employers [40,42] and were inadequately remunerated [40]. The literature also revealed that during the pandemic, some employers also restricted the movement of migrant farm workers and their communication with local support workers [40,42]. Some employers also reportedly inserted additional clauses into the employment contracts pertaining to curfews and not being able to leave the farm during their contracted period, while others deployed security officers to undertake excessive surveillance of workers [42]. While some employers attempted to justify these actions as a response to the pandemic, findings demonstrate that many farms actually performed poorly in implementing key COVID-19 protection measures (e.g., using crowded accommodations as self-isolation and quarantine facilities) [40,42]. Some employers also underpaid workers during quarantine periods or considered quarantine payments as a ‘loan’ despite the agriculture sector receiving nearly CAD 1 billion in government support pre-March 2020 to assist with increased operational costs and provide financial security for their workers [40]. Overall, both studies on migrant farm workers in Canada demonstrate that exploitative employment practices prevented effective implementation of COVID-19 measures, compromising migrant farm workers’ health and safety, and limiting their access to social protection.

For international students in Canada [44] and Australia [22], many lost their employment or had their work hours significantly reduced as a result of the pandemic. Although Canada and New Zealand have similar wage subsidy schemes that include temporary migrant workers [43,44], a significant number of international students in Canada still suffered financial hardship [44]. For those international students in Canada and Australia who lived precariously and relied on being able to work to meet their needs, loss of income resulted in difficulties in meeting basic housing and food needs [42,44].

The included studies further identified that non-state actors emerged as important sources of support and belonging for international students. Berg and Farbenblum [22] noted some educational providers, particularly universities, provided some financial support to students. NGOs also filled a significant gap in providing emergency food and clothing to international students [22]. Membership to workers unions was identified as another protective factor against job loss or reduced work hours for some international students [22].

### 3.5. Racism

Three of the included studies highlighted experiences of racism among migrant farm workers in Canada [40,42] and international students in Australia [22]. Racism itself is defined as “organized systems within societies that cause avoidable and unfair inequalities in power, resources, capacities and opportunities across racial or ethnic groups” [46] (p. 2).

For migrant farm workers in Canada, racism occurred within the context of exploitative employment practices, with sources identifying employers’ use of racial slurs and threats of deportation [36]. In one farming region, the local health authorities also mandated that migrant farm workers wear identification that provided proof of having completed quarantine—a practice argued to be akin to racial profiling [40].

Berg and Farbenblum [22] reported many Asian-born international students in Australia experienced verbal and physical abuse from people who accused them of ‘bringing’ COVID-19 into the country. This study also found those from African backgrounds experienced verbal abuse for ‘contributing’ to the spread of COVID-19. Berg and Farbenblum [22] further observed that racist incidents in Australia appeared to rise significantly after the Prime Minister, Scott Morrison, publicly announced that it was time for visitors and international students to go back to their home countries.

Similar to Australia, many migrant workers in New Zealand are also accused of contributing to the spread of COVID-19 and were told to go home to their ‘home country’ by other members of the public [43]. Overall, racism within political, community, and workplace settings perpetuated the sense of ‘othering’ and (non)belonging, further affirmed racism as a barrier to migrant workers accessing health services and social protection [40].

## 4. Discussion

To the authors’ knowledge, in the context of the COVID-19 pandemic, this is the first review examining access to health services and social protection among migrant workers in settler societies with comparable immigration, health, and welfare policies. This review contributes to the body of knowledge on the exclusion of migrants from the health system [13,17,47,48,49]. More specifically, this exploratory review provides contextualized understanding of access to health promoting resources across health services and social protection that are rarely examined together.

The findings shows that temporary migrant workers in Australia, Canada, and New Zealand remain largely excluded from health services and social protection, despite the inclusive COVID-19-related policies. The vulnerability to COVID-19 among temporary migrant workers, particularly migrant farm workers and international students, is magnified by a range of factors, including: (1) exacerbation of exploitative employment practices that contribute to poor implementation of COVID-19 measures and lack of support in accessing COVID-19 testing and treatment; (2) inadequate consideration of how precarity influences access to COVID-19 wage subsidies; and (3) racism that was perpetuated by politicians, employers, and members of the public [22,40,42,43,44]. Together, these factors reflect a dominant neoliberal agenda [23,24,50] and colonial histories that continue to shape immigration policies of these settler societies [18,19].

Figure 2 below provides visual representation of the interplay between three key concepts drawn from the included studies: exploitation, precarity, and racism. Based on the findings of this review, temporary migrant workers’ position in the labor market is governed by structurally racist practices and neoliberalism principles that tend to situate them as voiceless assets, prone to exploited, but not granted equitable access to health-promoting resources [40,51,52]. This form of systemic exploitation is practiced both explicitly and implicitly by employers and supported through legal and economic infrastructure developed and maintained by host governments [52,53]. Largely market-driven interactions between private sector and government actors also contributes to the precarity of temporary migrant workers by curtailing access to available social protection and maintaining high levels of employment insecurities [51,53]. Subsequently, the more precarious the workers’ living and working conditions, the more vulnerable they are to workplace exploitation [45]. This cycle of exploitation and precarity is heavily influenced by the institutions determined and maintained by host countries. As the primary providers of rights and resources, the actions and institutions of host countries may exclude individuals without formal citizenships from exercising their rights [54]. In the context of the COVID-19 pandemic, health and social policies responses for migrant workers may, hence, act solely as political symbols of inclusion when they are not supported by meaningful investments in community engagement and commitments to social justice.

Funding for services also plays a major role in providing support and resources for migrant workers [40,47]. However, even when funding exists, racism still acts as a major barrier for many migrant workers to access services that they are entitled to in a timely manner [48,55]. Findings of this review continue to demonstrate how racism governs many different settings and relationships within contemporary settler societies. Such structural racism reinforcing and, through embedded practice, seeks to normalize inequities experienced by racialized communities, including temporary migrant workers (see Figure 2) [12]. Structural racism enables exploitation of temporary migrant workers by maintaining environments where racist and othering practices are accepted and the needs and voices of temporary migrant workers are silenced or ignored [40,52]. This structurally embedded racism, alongside its more overt forms, also erodes institutional trust towards services and service providers among such ‘othered’ populations [40,52]. To understand the strategies employed by migrant workers to navigate structural racism in their everyday life, we need to consider the intersections of race with other social identities that construct migrant workers’ social position in the host countries [56]. For an example, the high representation of female Asian-born migrant workers in the included papers [22,43,44] indicates the need to examine how the intersections of race, ethnicity, gender, class, and immigration status influence access to health services and social protection [57,58].

Mediating some impacts of these exclusionary practices, NGOs emerged as key actors that provided information and services during the pandemic [22,40]. The Canadian government has responded to this by channeling investment for migrant farm workers through NGO outreach programs [34]. In this context, future research should also focus on community engagement to better mobilize information and support, as well as to reduce the impacts of structural racism [59].

### Limitations

This exploratory scoping review highlights and insufficient policy and research focus on the experience of migrant workers in the wake of COVID-19 in Australia, Canada, and New Zealand. This finding, however, is also a limitation of the study itself given the small number of included sources and the heterogeneity among the studies. Building on this foundation study, further research on this topic will be advantageous in the coming years as more detail on migrant workers’ experiences during the COVID-19 pandemic comes to light. Moreover, as de Haas, et al. [60] noted, migrants are defined and categorized in diverse ways and our definition of migrant workers may have some limitations. Furthermore, the study’s choice of databases and grey literature repositories may have inadvertently limited the identification of relevant papers. Misinterpretation of findings is also possible; however, robust discussion between authors and iterative analysis was embedded to minimize this.

## 5. Conclusions

Migrant workers account for a small portion of workers globally, but they have been disproportionately affected by the COVID-19 pandemic. Despite the enactment of a broad range of COVID-19-related policies in Australia, Canada, and New Zealand, many temporary migrant workers, such as migrant farm workers and international students, remained excluded from accessing health services and social protection. Three major factors play an important role in their exclusion: exploitation, precarity, and racism. Chief among these is structural racism. Born from their colonial histories and reimagined through the eyes of neoliberalism, the presence of structural racism in Australia, Canada, and New Zealand continues to privilege the economic use of migrants in building the wealth of host countries over a universalist approach to social justice. The COVID-19 pandemic has seen the functions fulfilled by many migrant workers recognized as essential by host countries. Moving forward, it is time we shed the market-driven basis for this ‘essential’ status and instead recognize the far broader contributions made by migrant workers to their host countries. This begins with more progressive, committed, and genuine action to recognize and address the structural determinants responsible for the health and welfare of migrants.

## Figures and Tables

**Figure 1 ijerph-19-02981-f001:**
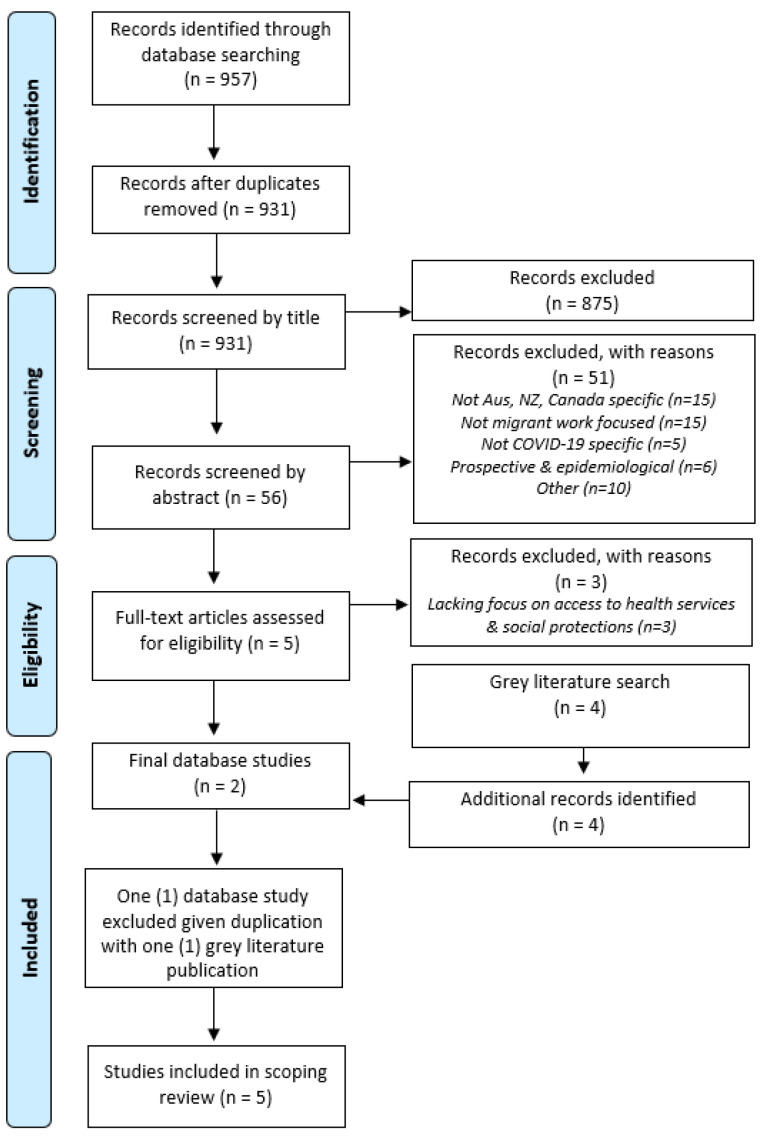
PRISMA diagram.

**Figure 2 ijerph-19-02981-f002:**
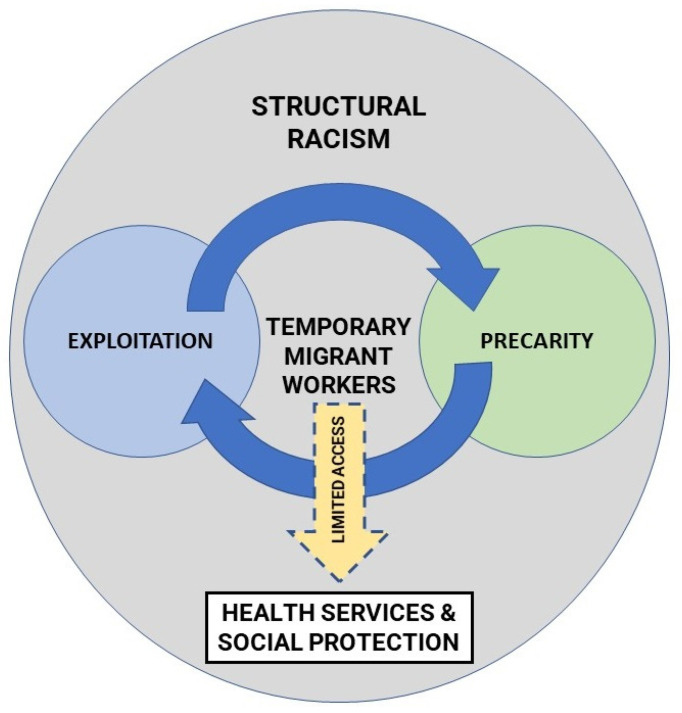
The interplay between exploitation, precarity, and structural racism and its impact on temporary migrant workers’ access to health services and social protection.

**Table 1 ijerph-19-02981-t001:** Search strategy ^1^.

Database	Search Terms	Records Obtained
Medline	**COVID**: COVID-19 or coronavirus or 2019-ncov or SARS-CoV-2 or COVID-19 **Immigrant:** Immigrant OR refugee OR migrant OR temporary resident **Risk**: risk OR vulnerable OR vulnerability **Location**: (Australia or Australian or Australians) OR (New Zealand or Aotearoa or NZ) OR (Canada or Canadian or Canadians)	28
PubMed		28
Scopus		901
Total		957

^1^ Search date: 17 December 2021.

**Table 2 ijerph-19-02981-t002:** Demographics of participants in the included studies.

Study	Country	Participant Demographics
Caxaj and Cohen [40]	Canada	30 individuals in support roles for migrant farm workers in British Columbia.
Migrant Workers Alliance for Change [42]	Canada	180 migrant farm workers who called a support hotline on behalf of 1162 workers.
Berg and Farbenblum [22]	Australia	6105 temporary migrant workers 52% aged ≥ 25 71% from Asian countries 54% female 51% had stayed for ≥18 months
Belong Aotearoa [43]	New Zealand	160 participants 81% aged ≥ 30 74% from Asian countries 69% female 51% arrived in the last 4 years 57% were temporary migrant workers
World Education Services [44]	Canada	4932 participants 90% aged ≥ 25 45% from India 54% female 33% had stayed for ≥12 months 52% were temporary migrant workers

## Data Availability

Not applicable.

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
