# Peer review of "(Not That) Essential: A Scoping Review of Migrant Workers’ Access to Health Services and Social Protection during the COVID-19 Pandemic in Australia, Canada, and New Zealand"

_ijerph, 2022, doi:10.3390/ijerph19052981_

Round 1

Reviewer 1 Report

There will be useful to add some chart and figures in order to highlight the results.

Reviewer 2 Report

Under the COVID-19 pandemic context, the migrant workers’ access to health services and social protection during the pandemic has become more challenging. This issue lies not only in the different public policies implemented by countries, but also is influenced in historic concepts or framework about the migrant workers’ rights in each country. The imbalance among the market competitions, government functions and public welfare and charitable organizations may lead to large-scale difference in integrating the migrant workers in the social security system. Thus, the review analysis could be multidisciplinary, covering sociology, political sciences, economical sciences and etc. This review focuses on recent policies and practices of migrant workers’ access to health service and social protection, and summarizes the development characteristics of the current situation in the settler societies with similar systems like Australia, Canada and New Zealand.

First and foremost, the review has reviewed a small number of studies about the implemented policies of countries and demonstrated the practices and comparisons about the migrant workers’ situation since COVID-19 context in the mentioned countries. Nevertheless, linked with the limitations of the actual policies or practices, not only the policy instrument perspective could be one important angle to deepen the understanding of this phenomenon, but also other social factors could be important to analyze this issue, such as political angle and from the social perspective. This nuanced and deeper difference from other perspectives could be mentioned or discussed.

Moreover, this review analysis needs more theoretical significance. For one thing, it needs to explore the theoretical perspective to study the migrant workers’ integration in settler societies in an evolving context by expanding the sources of studies in former studies. For another, the contextualization of migrant workers’ rights may be used to evaluate and compare the new trends in updated studies. That said, there are still several issues, both theoretical and empirical, in the article. I hope my comments can help the author(s) to improve the article.

Furthermore, according to the analytical framework, the discussions around the policy formulations or actions to improve the migrant workers have been partly mentioned. However, more theoretical discussions under the evolved context may concentrate more directly where the blind spots or neglected points stay in the policy settings, not only the policy instruments but also the essential concepts or state-society relations. This review should incorporate this insight in discussing the theoretical framework of protecting the migrant workers in an evolved context as well.  Without going into the details of the analysis, I would recommend the authors to revise and update the framework accordingly. 

Reviewer 3 Report

The authors cover an important area, that of disparities in access to health services and social protection during the COVID pandemic for migrant workers.  Although it is a very worthy topic, I have a few concerns:

  1. Although the authors cite that there is a lack of commitment from host countries (line 34), they also say that the WHO reviewed policies and that many high income countries have enacted inclusive policies (lines 38-41).  The authors did not mention if this included Canada, Australia, and New Zealand.
  2. Please define a settler society (line 56).  I wasn't quite sure why the three countries were included and others were excluded.
  3. In fact, there is heterogeneity within the 3 countries described (lines 102-122), so again, I'm not sure why these 3 countries were included and others excluded. It also makes for some difficulty in making generalizations.  
  4. In the screening process, the authors should describe the role of the authors a bit more (lines 151-152).
  5. The finding that exploration of access of migrant workers remains underexamined due to the dearth of literature is not surprising as the pandemic is not yet 2 years old, and therefore, much of the literature has not yet been published on any population. (lines 208-209)
  6. The sections on racism are particularly important and well done...the examples are striking (but sadly, not surprising). (lines 246-249, 255-257)
  7. The sections discussing lack of social protection also have good examples that highlight the disparities (lines 272-278).
  8. Because of the heterogeneity of the countries, their policies, and the populations described in the five studies, although I think this is an important topic, there just doesn't seem to be enough there to make strong conclusions.  In addition, the authors themselves state that the findings are similar to other studies (lines 332-333), so I'm not sure what more this has to add (at this time).

My recommendation would be to repeat this study in a few years when there is more literature to review.

Round 2

Reviewer 1 Report

It is better now

Reviewer 2 Report

The revised version has been much improved according to the reviewers' comments.

Reviewer 3 Report

Thank you for your cover letter describing your revisions and for the revisions themselves.  The revisions that you made have resulted in a paper that, though, I believe, still preliminary due to literature related to this topic still being limited due to the recent onset of COVID, reads much better and provides information that can be used as a starting place for more robust work.  I appreciate you addressing the definitions (e.g., settler), the inconsistencies with the three countries, and more specifically addressing the process.  I have no specific revisions at this time as you addressed my concerns.